# Growth and Nutrient Element Content of Hydroponic Lettuce are Modified by LED Continuous Lighting of Different Intensities and Spectral Qualities

**Wenke Liu** [1,2,*,†] , **Lingyan Zha** [1,2,†] and **Yubin Zhang** [1,2]

1   Institute of Environment and Sustainable Development in Agriculture, Chinese Academy of Agricultural Sciences, Beijing 100081, China; zhaly2013@163.com (L.Z.); zhangyubin26@163.com (Y.Z.)
2   Key Lab of Energy Conservation and Waste Management of Agricultural Structures, Ministry of Agriculture and Rural Affairs, Beijing 100081, China
*   Correspondence: liuwenke@caas.cn
†   Both authors contributed equally to this work.

**Abstract:** **LED** red (R) and blue (B) continuous light (CL) is a potential efficient way to increase plant productivity of plant factory with artificial light (PFAL), but limited information was explored about their effects on plant mineral nutrition. In an environmentally controlled plant factory with artificial light (PFAL), the effects of CL of different intensities and spectral qualities, emitted by R and B LEDs on growth and nutrient element content and accumulation of lettuce (*Lactuca sativa* L.), were conducted in three hydroponic experiments. Two treatments, normal light (12 h/12 h) and CL (24 h/0 h) in experiment 1, three CL intensities (100, 200 and 300 μmol·m$^{-2}$·s$^{-1}$) in experiment 2, and three CL light qualities (1R:3B, 1R:1B and 3R:1B) in experiment 3 were designed. The results showed that CL significantly increased the fresh and dry lettuce shoot biomass compared with normal light, and shoot fresh and dry biomass increased with the intensity increment of CL. In experiment 3, shoot fresh biomass was great under high R light proportion CL treatment, while dry shoot biomass remained unchanged. Both CL and CL with increased intensities promoted shoot C content and accumulation in lettuce. CL reduced N, P, K, Ca, Mg, Cu and Zn contents in lettuce shoot, while Fe and Mn contents did not change compared to NL. Moreover, CL increased Ca, Fe and Mn accumulation. 100–200 μmol·m$^{-2}$·s$^{-1}$ CL facilitated N, P, Ca, Mg, Fe, Mn, Cu and Zn contents in shoot, but K content was not influenced compared with 300 μmol·m$^{-2}$·s$^{-1}$. The data showed that high B light ratio (75%) facilitated C content comparison with low B ratios (50% and 25%). However, lettuce grown under 3R1B treatment had the higher C accumulation. Shoot N, P, K, Ca, Mg, Fe, Mn and Zn contents were higher under 1R1B treatment, and Cu content did not affected by light quality. Moreover, accumulation of N, P, K, Ca, Mg, Fe, Mn, Zn and Cu in shoot was higher under 1R1B treatment, while P, Ca, Mg, Mn accumulation under 3R1B treatment was the lowest. In conclusion, CL tends to reduce shoot mineral element contents due to dilution effect as shoot dry weight increases compared to NL. However, long-term (12 days) CL composed of 1R1B, 100–200 μmol·m$^{-2}$·s$^{-1}$ tends to obtain relative higher K, Ca, Fe and Zn contents in the greater dry lettuce shoot.

**Keywords:** continuous light; hydroponic lettuce; light intensity; light quality; nutrient uptake

## 1. Introduction

Until now, many crops, mainly horticultural crops (e.g., leaf vegetable, root vegetable, and fruit vegetable) and some medicinal plants (glycyrrhiza, ginseng, mint etc.), have been successfully cultivated via hydroponics or substrate cultivation, such as light-emitting diode (LED), in a closed growth chamber [1,2] or plant factory with artificial light (PFAL). Productivity of PFAL is high because almost

all inner environmental factor-related plant growth is controlled integrally according to physiological requirements. Nowadays, PFAL has become an important indoor agricultural system to produce vegetables, medicinal plants and seedlings in large scales, particularly in East Asia countries, including China, Japan and Korea etc. [3]. Moreover, PFAL has more potential to enhance its productivity through LED light environmental control, establishing specific light modes by integrating spectral quality, intensity, photoperiod, and circadian rhythm traits at microsecond to hour levels [4,5]. Taking advantage of spectral modulability and intelligent control of LED lamps, many studies indicated that light intensity, quality and photoperiod even lighting mode regulation affectively altered horticultural crop growth, yield and even nutritional quality [2,4,6–8]. Therefore, some meaningful lighting methods for PFAL, e.g., continuous light (CL), alternating light, etc., were recommended for purposely improving for high yield or health-beneficial phytochemicals accumulation without extra energy input.

Many reports had indicated that CL with LED light sources could improve yield and nutritional quality before harvest or during long-term cultivation. Some studies showed that 24 h–72 h CL with combinative light of red (R) and blue (B), or R, B and green light enhanced soluble sugar and vitamin C contents while reduced nitrate content in hydroponic lettuce (*Lactuca sativa* L.) [9–11]. Overall, lettuce plants could gain more productivity and grow normally without leaf injury under 200 $\mu mol \cdot m^{-2} \cdot s^{-1}$ red and blue CL during 15 days, while the most cost-effective duration of CL for yield improvement was nine days [12]. Our previous data showed that lettuce plants gained greater shoot fresh weight and dry matter under long-term CL (200 $\mu mol \cdot m^{-2} \cdot s^{-1}$) after 15 days of growth in comparison with normal photoperiod (16/8 h, 200 $\mu mol \cdot m^{-2} \cdot s^{-1}$) [12]. Moreover, both ascorbate content and dehydroascorbate content were remarkably enhanced by CL, and remained a relatively stable level during the experiment [12]. Therefore, PFAL can be used to produce high-quality leafy vegetables with more biomass and nutritional substance (e.g., antioxidants, mineral elements) accumulation under specific CL conditions.

Plants do need seventeen nutrient elements at least, and some are mineral elements with important physiological functions both, for plant growth and human health. It was reported that some plant necessary nutrient elements (such as zinc, iron, calcium, potassium etc.) and non-essential health-beneficial mineral element (e.g., selenium) might be absorbed more by vegetables grown in PFAL via light environmental regulation, particularly preharvest CL [13,14]. How can the content be regulated or these beneficial mineral elements be accumulated is still an urgent issue for PFAL, in order to cater the increasing needs of healthy diet. However, how CL with various monochromatic light combinations affects mineral element content is still an issue without fully investigation. Nowadays, some useful information about spectral quality, and temporal shift of light quality on mineral element contents in leafy vegetables had been reported [15–19]. Kopsell et al. (2013) [15] found that many nutrient element contents of broccoli seedlings increased significantly after 5 days shift of light environment from 350 $\mu mol \cdot m^{-2} \cdot s^{-1}$ red and blue light (12%B and 88%R) to 41 $\mu mol \cdot m^{-2} \cdot s^{-1}$ pure blue light. Kopsell et al. (2014) [16] showed that many nutrient element contents of broccoli seedlings exposed to high blue light ratio lighting (20%B and 80%R; 20%B, 70%R, and 10% green light) were greater than those under incandescent lamp and fluorescent lamp of the same intensity (250 $\mu mol \cdot m^{-2} \cdot s^{-1}$). Moreover, nutrient element contents of lettuce under fluorescent lamp and different combinations of R and B lights were investigated [17,18]. Further, Chen et al. (2015) [19] investigated the effects of five light sources, fluorescent lamp, fluorescent lamp+LED red and blue light, LED red and blue light and monochromatic red light and blue light, on nutrient element contents for hydroponic *Taraxacum mongolicum*.

Lettuce is one of the most popular leafy vegetables which usually cultivated hydroponically worldwide, particularly in PFAL. In fact, health-beneficial mineral elements, such as Ca, Fe, Zn and Se and so on, are desired to improve contents or accumulation in hydroponic lettuce through light environmental control. Therefore, the effects of light intensity and spectral quality of CL emitted by red and blue LEDs on growth and nutrient elements uptake were conducted in three experiments using ICP-AES technology determination. The objectives of this study are to clarify the effects of red and blue CL on contents and accumulation of ten nutrient elements, and confirm intensity and quality effects

of red and blue CL on contents and accumulation of ten nutrient elements. The results will provide useful data for PFAL to produce lettuce with high health-beneficial mineral elements under CL.

## 2. Materials and Methods

### 2.1. Plant Materials and Growth Conditions

The experimental trials were carried out in an environment-controlled plant factory at 23 ± 3 °C, HR 50–60%. Lettuce cultivar used in present studies was "Yidali", and the lettuce seeds were purchased from the market. Three experiments were included in present studies, the first one was carried out during April to May, 2018, and other two experiments were simultaneously conducted during July to August, 2018. The seedling conditions of the three experiments were completely consistent. Lettuce seeds were sown in sponge cubes ($2.5 \times 2.5 \times 2.5$ cm) (purchased from Zhonggengmuming Agricultural Technology Development Co. Ltd., Beijing, China) and germinated under 200 $\mu mol \cdot m^{-2} \cdot s^{-1}$ (16 h/8 h, light/dark) irradiance provided by white LED panel ($50 \times 50$ cm, Shenzhen Huihao Optoelectronic Co. Ltd., Shenzhen, P. R. China) for 15 days in an environment-controlled plant factory. Environmental conditions in the plant factory were set at 23 ± 2 °C, 60% relative humidity (RH) and atmospheric $CO_2$ concentration. After the second leaf was fully expanded, the seedlings were randomly transplanted into hydroponic pots ($50 \times 50 \times 6$ cm). Each pot can contain 13 plants and 10 L nutrient solution. The nutrient solution composition was as follow (mmol/L): 0.75 $K_2SO_4$, 0.5 $KH_2PO_4$, 0.1 KCl, 0.65 $MgSO_4 \cdot 7H_2O$, $1.0 \times 10^{-3}$ $H_3BO_3$, $1.0 \times 10^{-3}$ $MnSO_4 \cdot H_2O$, $1.0 \times 10^{-4}$ $CuSO_4 \cdot 5H_2O$, $1.0 \times 10^{-3}$ $ZnSO_4 \cdot 7H_2O$, $5 \times 10^{-6}$ $(NH_4)_6Mo_7O_{24} \cdot 4H_2O$, 0.1 EDTA-Fe, 4 $Ca(NO_3)_2 \cdot 4H_2O$ (pH: 5.90, EC: 1.24 $mS \cdot cm^{-1}$). To maintain the EC and pH, the nutrient solution was completely replaced each week.

### 2.2. Light Treatments of Three Experiments

After transplanting, lettuce seedlings in each pot were irradiated by a red (R) and blue (B) LED light panel with the peak wavelengths of 655 nm and 430 nm. The details of LED fixtures used in the three experiments are described as Zha et al. (2019) [20]. During the first 7 days (experiment 1) or 10 days (experiment 2 and 3) after transplanting, all lettuce seedlings were irradiated with the same light condition (3R:1B, 200 $\mu mol \cdot m^{-2} \cdot s^{-1}$ and 16 h/8 h). Then, seedlings were randomly divided into various groups for different light treatments, and each treatment contains three pots seedlings (i.e., 39 plants). Experiment 1 included two light treatments, i.e., normal light (NL, 16 h/8 h) and CL (24 h/0 h) with light quality 3R:1B and 200 $\mu mol \cdot m^{-2} \cdot s^{-1}$ light intensity. Experiment 2 contained three CL treatments with different light intensities (100, 200 and 300 $\mu mol \cdot m^{-2} \cdot s^{-1}$) and same light quality (3R:1B). Experiment 3 contained three CL treatments with different light qualities (1R:3B, 1R:1B and 3R:1B) and same light intensity (200 $\mu mol \cdot m^{-2} \cdot s^{-1}$) (Figure 1). The light treatment details for three experiments were listed in Table 1. Light intensity was measured at the canopy level by a light sensor logger (Li-1500; LI-Cor, Lincoln, NE, USA), then adjusted to set values.

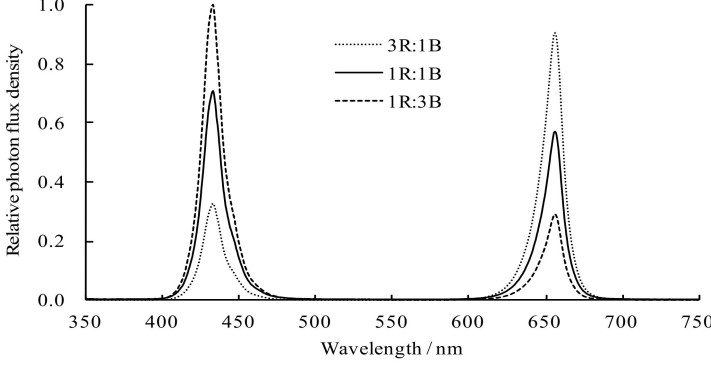

**Figure 1.** Spectral distribution of LED lights with different red:blue ratio.

**Table 1.** The details of light treatments for three experiments.

| Experiments Number | Treatment Number | Photoperiod (h) | Light Intensity /(μmol·m$^{-2}$·s$^{-1}$) | Red to Blue Light Ratio |
|---|---|---|---|---|
| 1 | 1 (NL) | 16/8 | 200 | 3R:1B |
| | 2 (CL) | 24/0 | 200 | 3R:1B |
| 2 | 1 | 24/0 | 100 | 3R:1B |
| | 2 | 24/0 | 200 | 3R:1B |
| | 3 | 24/0 | 300 | 3R:1B |
| 3 | 1 | 24/0 | 200 | 1R:3B |
| | 2 | 24/0 | 200 | 1R:1B |
| | 3 | 24/0 | 200 | 3R:1B |

### 2.3. Sampling and Measurement Methods

Lettuce plants of three experiments were treated for 12 days. Then, three plants in each treatment were sampled randomly for determination. After weighing shoot fresh weight, seedlings were then dried under 80 °C in oven for three days. Dry weights were then determined. After grinding, 1 g shoot dry sample was weighed for cook elimination by adding perchloric acid and concentrated nitric acid at 180 °C. Atomic absorption spectrophotometer and inductively coupled plasma source mass spectrometer were used to determine ten nutrient element contents, i.e., carbon (C), nitrogen (N), phosphorus (P), potassium (K), calcium (Ca), magnesium (Mg), iron (Fe), manganese (Mn), copper (Cu) and zinc (Zn). The accumulation of all nutrient elements in the lettuce shoot was calculated by nutrient element content multiply by dry shoot weight.

### 2.4. Statistical Analysis

Data were analyzed by the statistical software SPSS 18.0 (International Business Machines Corporation). Significant differences between different light treatments were tested by Tukey test at 95% confidence.

## 3. Results

### 3.1. Lettuce Biomass

Table 2 shows that CL in experiment 1 significantly increased shoot fresh and dry weight compared with NL. Also, CL with increased intensity remarkably improved shoot fresh and dry weight compared with CL treatment with 300 μmol·m$^{-2}$·s$^{-1}$ light intensity in experiment 2. However, with red light proportion increment (25% to 75%) in CL spectral quality, shoot fresh biomass increased gradually while dry shoot biomass was unchanged in experiment 3.

**Table 2.** Effects of light intensity and quality of continuous lighting emitted by red/blue LEDs on shoot biomass of lettuce.

| Experiments Numbers | Treatment Number | Treatment | Shoot Fresh Weight/g | Shoot Dry Weight/g |
|---|---|---|---|---|
| 1 | 1 | NL | 41.5 ± 4.2b | 2.23 ± 0.15b |
| | 2 | CL | 48.9 ± 3.9a | 2.90 ± 0.34a |
| 2 | 1 | 100 | 76.9 ± 2.5c | 2.06 ± 0.35b |
| | 2 | 200 | 98.8 ± 6.7b | 3.48 ± 0.53a |
| | 3 | 300 | 109.9 ± 4.3a | 4.16 ± 0.35a |
| 3 | 1 | 1R:3B | 88.5 ± 3.2b | 2.98 ± 0.42a |
| | 2 | 1R:1B | 94.2 ± 3.1ab | 3.02 ± 0.32a |
| | 3 | 3R:1B | 99.5 ± 4.3a | 3.55 ± 0.42a |

Note: Different letters in the same column indicate a significant difference among treatments at *p* < 0.05.

### 3.2. Nutrient Element Contents and Accumulation of Lettuce under NL and CL

There are large differences in nutrient element content and accumulation between CL and NL treatments (Figure 2). CL significantly decreased N, P, K, Ca, Mg, Cu and Zn contents in lettuce shoot, and increased C content, while Fe and Mn contents remained unchanged. CL significantly increased C, Ca, Fe and Mn accumulation in lettuce shoot, while N, P, K, Mg, Cu and Zn contents remained unchanged.

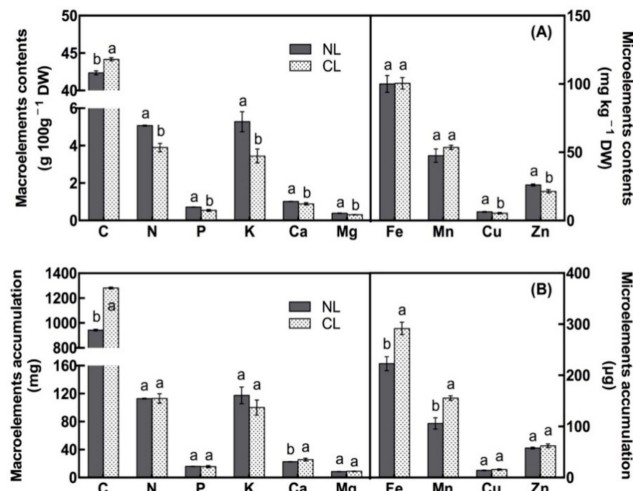

**Figure 2.** Effects of continuous light and normal light on nutrient element contents; (**A**) and accumulation; (**B**) of hydroponic lettuce. Different letters indicate a significant difference among treatments at $p < 0.05$.

### 3.3. Nutrient Element Contents and Accumulation of Lettuce under CL of Different Intensities

CL with increased intensities increased C content of lettuce shoot, and N, P, Ca, Mg, Fe, Mn, Cu and Zn contents in lettuce shoot decreased significantly, while K content remained unchanged (Figure 3). Meanwhile, CL with increased intensities increased all nutrient element accumulation except Mn and Cu. Moreover, under CL treatments with 200 and 300 μmol·m$^{-2}$·s$^{-1}$, contents and accumulation of more than half nutrient elements in lettuce shoot were similar with no significant difference. Therefore, 200 μmol·m$^{-2}$·s$^{-1}$ is recommended as the potent light intensity for LED red and blue CL.

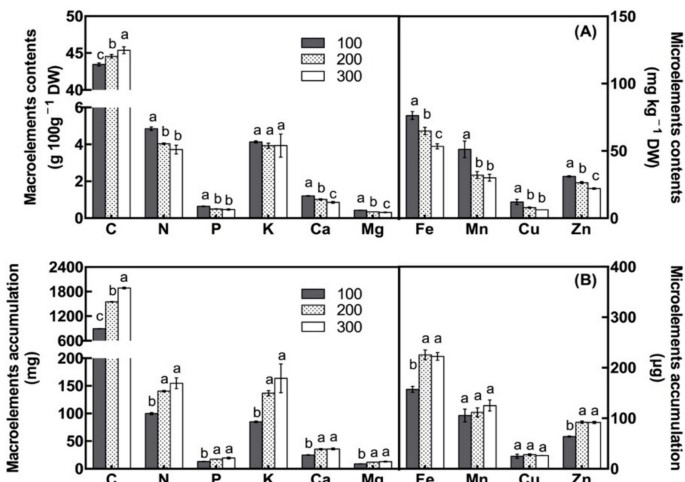

**Figure 3.** Effects of continuous light with different light intensities on nutrient element contents; (**A**) and accumulation; (**B**) of hydroponic lettuce. Different letters indicate a significant difference among treatments at $p < 0.05$.

### 3.4. Nutrient Element Contents and Accumulation of Lettuce under CL of Different Spectral Qualities

The effects of CL spectral quality on nutrient element content and accumulation are more complex than light intensity (Figure 4), and nutrient element contents are spectral quality depended. For carbon content in lettuce shoot, lettuce plants grown under 1R1B had the lowest carbon content, and that under 1R3B presented the highest content compared with other light qualities. The data means that high blue light ratio (75%) facilitated carbon content comparison with low ratios (50% and 25%). However, shoot carbon accumulation in lettuce grown under 3R1B treatment had higher carbon accumulation. Apparently, carbon accumulation depended on shoot dry mass and carbon content in it. Contents of N, P, K, Ca, Mg, Fe, Mn and Zn in shoot were higher under 1R1B treatment, and Cu content did not affected by light quality. Moreover, accumulation of N, P, K, Ca, Mg, Fe, Mn, Zn and Cu in shoot was higher under 1R1B treatment, and accumulation of P, Ca, Mg, Mn of 3R1B treatment was the lowest among three light qualities. In addition, accumulation of N, K, Fe, Zn and Cu was similar for three light qualities.

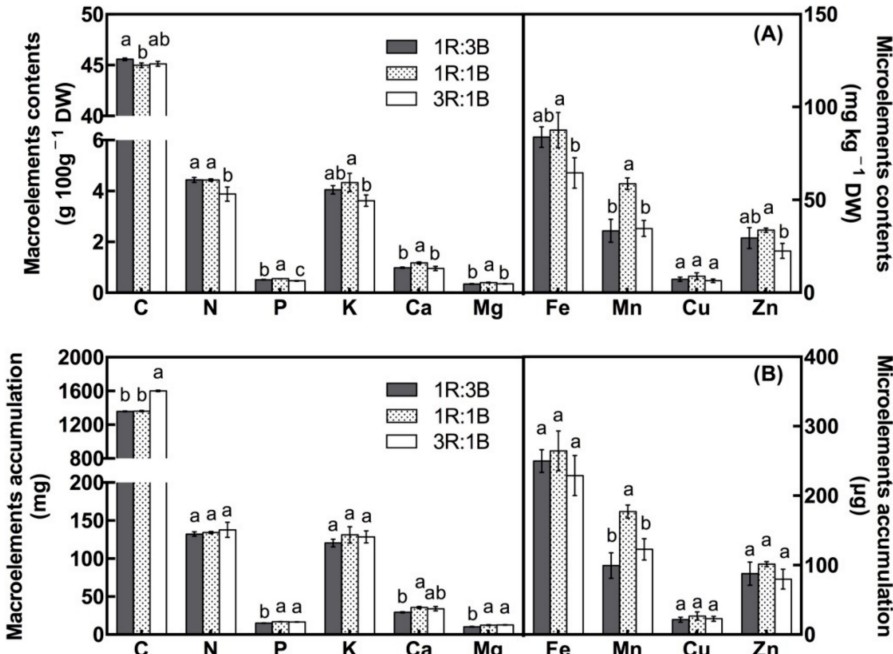

**Figure 4.** Effects of continuous light of different light qualities on nutrient element contents; (**A**) and accumulation; (**B**) of hydroponic lettuce. Different letters indicate a significant difference among treatments at $p < 0.05$.

### 4. Discussion

The current data showed that hydroponic lettuce grown under red and blue CL for 12 days could gain higher shoot fresh and dry weight simultaneously, also lettuce grown under higher CL intensity gained greater shoot fresh and dry biomass further. Moreover, CL spectral quality did alter shoot fresh weight, but not dry weight, low blue light ratio favored fresh weight accumulation of lettuce. Apparently, up-regulation of lettuce fresh yield and dry biomass accumulation by CL was due to prolonged photosynthetic duration, compared with NL. In addition, high-intensity CL had further improved lettuce fresh yield and dry biomass accumulation, compared with low-intensity CL, which might be due to higher net photosynthetic rate under higher light intensity. High blue light ratio in CL reduced lettuce fresh yield, which might result from leafy physiological changes caused by blue light. First, efficiency of blue light in regulating stomatal opening was almost ten times of red light [21]. Second, water retention ability of leafy stomata exposed to CL was lower than that to

NL [22]. Therefore, shoot water content of lettuce grown under CL is usually lower than that of lettuce grown under NL.

Compared with NL, CL including increased C content, reduced N, P, K, Ca, Mg, Cu and Zn contents, while unchanged Fe and Mn contents of lettuce. The result indicated that lettuce shoot accumulated more carbon under CL through improved photosynthetic capacity. Higher carbon content in lettuce shoot might result in lower N, P, K, Ca, Mg, Cu and Zn contents due to dilution effect if their absorption rates did not accelerate under CL to match C accumulation rate. Therefore, Fe and Mn contents of lettuce shoot under CL did not change which might result from the offset of increased absorption quantity and dilution effect caused by carbon accumulation. Eventually, the accumulation of C, Ca, Fe and Mn of lettuce shoot were increased under CL due to high contents and great shoot dry biomass. We confirmed that lettuce shoot could accumulate more Ca, Fe and Mn when provided long-term 24 h photoperiod.

CL of 100 and 200 $\mu mol \cdot m^{-2} \cdot s^{-1}$ facilitated the contents of N, P, Ca, Mg, Fe, Mn, Cu and Zn in shoot compared with 300 $\mu mol \cdot m^{-2} \cdot s^{-1}$, but K content was not influenced. The high intensity CL was in favor of element accumulation, and 200 $\mu mol \cdot m^{-2} \cdot s^{-1}$ is suitable and recommended. High light intensity CL could accumulate more carbon in lettuce shoot which might led to reduction in contents of N, P, Ca, Mg, Fe, Mn, Cu and Zn. Actually, nutrient element content and accumulation totally depend on photosynthesis and nutrient element absorption rates that are. Under certain DLI range, photosynthetic rate and nutrient absorption rate increase with light intensity increasing. Although, photosynthetic carbohydrates quantity in leaves under CL increased with elevated intensity, the transportation ratio of photosynthetic carbohydrates to roots became less due to the lack of darkness period. Therefore, the energy, carrier and carbon skeleton for nutrient absorption and assimilation was reduced substantially. Even so, our data showed that responses of nutrient element contents were inconsistent which means that there may exist some complex mechanisms that CL intensity dominate nutrient uptake.

The data means that high blue light ratio (75%) facilitated carbon content enhancement comparison with low B ratios (50% and 25%). However, shoot carbon accumulation in lettuce grown under 3R:1B treatment was the higher. Apparently, carbon accumulation depended on shoot dry mass and carbon content jointly. Contents of N, P, K, Ca, Mg, Fe, Mn and Zn in shoot were higher under 1R:1B treatment, and Cu content was not affected by light quality. Moreover, accumulation of N, P, K, Ca, Mg, Fe, Mn, Zn and Cu in shoot was higher under 1R:1B treatment, and accumulation of P, Ca, Mg, Mn under 3R:1B treatment was lowest under three light qualities. However, accumulation of N, K, Fe, Zn and Cu was similar for three light qualities. The present data indicated CL of 1R:1B could effective improve K, Ca, Fe and Zn contents in shoot, which is an important finding with great significance for high nutritional lettuce cultivation in PFAL. Ten nutrient element contents and accumulation were regulated by intensity and quality of CL, by which we can acquire commercial lettuce being rich in or lack of some special nutrient elements to cater the requirements of consumers. The optimal intensity and quality of CL for maximal shoot fresh and dry weight, and nutrient element content and accumulation of hydroponic lettuce were summarized in Table 3. Table 3 indicates that shoot fresh and dry weight, and nutrient element content and accumulation are dependent on both light intensity and quality. We suggested that low intensity (100 $\mu mol \cdot m^{-2} \cdot s^{-1}$) and medium 50% blue light is the best CL condition for high nutrient element contents, while high intensity (200–300 $\mu mol \cdot m^{-2} \cdot s^{-1}$) and medium to high (50–75%) blue light is the best CL condition for high nutrient element accumulation. Apparently, the later were codetermined by nutrient element content and dry shoot biomass jointly. Taking double time up-regulation of shoot dry weight and nutrient element contents by intensity and quality of CL into considerations, long-term CL composed of 1R1B, 100 $\mu mol \cdot m^{-2} \cdot s^{-1}$ for 12 days tends to increase K, Ca, Fe and Zn contents in shoot totally because CL reduced K, Ca and Zn content at small rate, and did not reduce Fe content. Overall, it is recommended that the yield and health-beneficial nutrient element contents in lettuce is increased by ensuring suitable intensity and quality, as well as duration of CL composed of red and blue light.

**Table 3.** Optimal intensity and quality of CL for maximal shoot fresh and dry weight, C and nutrient element content and accumulation of hydroponic lettuce.

| Light Parameters | Shoot Fresh Weight | Shoot Dry Weight | C Content | C Accumulation | Nutrient Element Content | Nutrient Element Accumulation |
|---|---|---|---|---|---|---|
| Light intensity ($\mu mol \cdot m^{-2} \cdot s^{-1}$) | 300 | 200–300 | 300 | 300 | 100 | 200–300 |
| Spectral quality (%, blue light) | 25 | 25–75 | 75 | 25 | 50 | 50–75 |

Spectral quality effects on nutrient element absorption and utilization of plants have not fully investigated and understood. Actually, both monochromatic red light or blue light is not suitable for hydroponic lettuce production in PFAL in terms of yield, thus the combinations of red and blue light are basic spectral quality for crop cultivation in PFAL and CL application. Many reports had evidenced that compound RB light performed better in cultivating crops [23]. The biomass production of both cultivars grown under RB conditions was higher than that of plants grown under R conditions. This enhancement of biomass production was caused by an increase in the net assimilation rate (NAR). The higher NAR was associated with a higher leaf N content per leaf area at the whole-plant level, which was accompanied by higher contents of the key components of photosynthesis, including Rubisco and chlorophyll [23–25]. Hogewoning et al. (2010) [26] suggested that the photosynthetic capacity was twice as high for leaves grown at 7% blue compared with 0% blue, and continued to increase with increasing blue percentage during growth measured up to 50% blue. The increase in maximum net photosynthetic rate with blue percentage (0–50%) was associated with an increase in leaf mass per unit leaf area, N content per area, chlorophyll content per area, and stomatal conductance.

LED is a promising technology with the potential to improve the irradiance efficiency, light quality, and the light spectrum for increasing plant yield and quality. Amoozgar et al. (2017) [27] found that LED light elevated the concentrations of macro-and micronutrients in lettuce possibly because of the direct effect of LED light and lower stress conditions in the growth chambers compared to the greenhouse. Although, the mechanism of the changes in lettuce grown under LED is not well-understood, the results of this study demonstrated that LED light could be used to enhance the growth and nutritional value of lettuce in indoor plant production facilities. Monostori et al. (2018) [28] conclude that compared to red and blue light, white light had generally similar effects on seedling growth at the same PPFD with similar electric energy consumption, and improved the visual color quality of sole-source lighting. Chen et al. (2014) [18] indicated that nutrient element contents for hydroponic lettuce were higher under monochromatic red light than those under RB light. However, accumulation of these nutrient elements was less due to their small biomass. In summary, productivity and mineral element contents of lettuce in plant factory can be regulated through CL and its light intensity and quality.

CL modified growth and metabolism of plants grown under LED light sources, which was affected by the intensity, spectral quality, and duration (long-term or short-term) of CL. Nowadays, there is still a lack of detailed information on the lighting conditions required for optimal growth of different plant species, and the effects of light intensity and spectral composition on plant metabolism and nutritional quality of LED monochromatic and combinative light. There were species-depended differences in the responses to CL and CL with various intensities and qualities, and the physiological mechanisms underlying is still needed to be investigated. Totally, nutrient element contents of lettuce depend on shoot dry mass, nutrient element absorption rate, and distribution in plants instead of transformation and assimilation. Carbon and nitrogen metabolism requirements for nutrient elements are influenced by blue and red light, which is the dominant power for nutrient element absorption rate change. It is proposed that long-term CL with proper intensity and quality is a potential way to produce nutritional lettuce, with high or low mineral element content, as desired. Further investigations should be conducted to improve the efficiency of CL process regulation by integrating special light modes reported [5,7,8].

## 5. Conclusions

Long-term RB CL (12d) significantly altered hydroponic lettuce yield and nutrient element content which depended on light intensity and quality. CL with increased light intensity improved the fresh and dry lettuce shoot biomass. Also, high R light proportion of CL spectral quality, shoot fresh biomass increased gradually, while dry shoot biomass was unchanged. Shoot C and nutrient element content, and the accumulation was dependent on both light intensity and quality. Both CL and CL with increased intensities promoted shoot carbon content and accumulation in lettuce. Moreover, CL reduced N, P, K, Ca, Mg, Cu and Zn contents in shoot, while Fe and Mn content did not change. Also, CL increased Ca, Fe and Mn accumulation. The contents of N, P, Ca, Mg, Fe, Mn, Cu and Zn in shoot was facilitated by 100–200 $\mu mol \cdot m^{-2} \cdot s^{-1}$ CL, but K was unchanged under 300 $\mu mol \cdot m^{-2} \cdot s^{-1}$. However, lettuce grown under 3R1B treatment had the higher carbon accumulation. Contents of N, P, K, Ca, Mg, Fe, Mn and Zn in shoot were higher under 1R1B treatment, and Cu content did not affect by light quality. Moreover, accumulation of N, P, K, Ca, Mg, Fe, Mn, Zn and Cu in shoot was higher under 1R1B treatment, and accumulation of P, Ca, Mg, Mn under 3R1B treatment was the lowest under three light qualities. In addition, the accumulation of N, K, Fe, Zn and Cu was similar for three light qualities. In conclusion, long-term CL with selected intensity and quality is a feasible method to improve plant productivity of PFAL in terms of yield and mineral element contents (e.g., K, Ca, Fe and Zn) of hydroponic lettuce. CL of 100–200 $\mu mol \cdot m^{-2} \cdot s^{-1}$ and 1R1B are better light condition for high beneficial element contents with economic electric input.

**Author Contributions:** W.L. provided equipment and funds, and conducted the experiment, together with L.Z., also wrote the entire manuscript. L.Z. and Y.Z. conceived the experiment, and prepared the materials for the experiment, collected samples, and undertook determination work. All authors have read and agreed to the published version of the manuscript.

**Funding:** This research was funded by the National Natural Science Foundation of China (NSFC) (grant No. 31672202).

**Acknowledgments:** This research was financially supported by the National Natural Science Foundation of China (NSFC) (grant No. 31672202).

**Conflicts of Interest:** The authors declare no conflict of interest.

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
