# Peer review of "Growth and Nutrient Element Content of Hydroponic Lettuce are Modified by LED Continuous Lighting of Different Intensities and Spectral Qualities"

_agronomy, doi:10.3390/agronomy10111678_

Round 1
Reviewer 1 Report
In the manuscript, actual and not yet widely analyzed question of light impact on lettuce mineral nutrition is explored. The results are of high interest for readers, however, the quality of presentation and deficient methodological details (see below) highly diminish the overall value of the manuscript. The experimental design and the methodological details of the manuscript overlap with the already published research of these authors "Morphological and Physiological Stress Responses of Lettuce to Different Intensities of Continuous Light" https://www.frontiersin.org/articles/10.3389/fpls.2019.01440/full
However this relation it is not explored in current manuscript.
TITLE - how authors according their results can separate the terms of mineral element "contents" and "accumulation"? As the process of accumulation was not evaluated, I would suggest to leave only "contents"
ABSTRACT - short, but methodical and result parts are very general. Blue/red light ratio expressed in %, when further in the manuscript - their ratio.
INTRODUCTION
There and elsewhere in the text - major revision of language style is necessary. The contents of introduction are suitable, however, literature references are >5 years old, and no scientific sources of recent years discussed. Part of information, suitable in introduction was explored in discussion section.
L83-L87 - it is more actual to know, what was determined, rather than what was investigated.
L88-92 - reference source for substantiation?
MATERIALS AND METHORD
L101 - latin name in italic
L102 - cultivar name should be presented in ""
L117-124 not clearly described.
Where the experiments repeated in time? or these are results of single replication? Experiments 1,2 and 3 were of different duration? How this could affect the results?
Manufacturers of analytical equipment not provided.
DLI (daily light integral) could be provided for lighting treatments.
RESULTS - authors only operate with terms "the highest" "the lowest, but the extent of the differences between treatments (in %, or times) is not presented.
DISCUSSION - is very long, consists of two parts. In the first one, the summary of results (also repeating the result information). Second part - Analysis of literature of the similar studies. Both parts should be mixed together to make the discussion more consistent. Most of the analysis of literature is more suitable for introduction rather than for discussion.
L270 - 271 Better than what?
L271 - 272 - is it a reference?
L276 - what plant?
L 278 - what is "Amax" ? What kind of plant?
L305-306 - is it a reference or a result>
CONCLUSIONS - too much of "result" style.
REFERNCES - should be revised for literature sources of last few years.
Author Response
Responses to reviewer 1
First of all, we want to thank the reviewer for your helpful comments. We had revised the manuscript thoroughly according to reviewer comments.
- The materials and methods section was revised, and necessary information was supplemented.
- The published paper "Morphological and Physiological Stress Responses of Lettuce to Different Intensities of Continuous Light" was cited.
- The title was changed by deleting the word "accumulation".
- Abstract is rewritten, and the introduction was revised carefully.
- All problems put forward in the comments for reviewer 1 were revised.
- Results and discussion were revised.
- Some new references were added.
Reviewer 2 Report
The study is actual and interesting. But substantial part of the Introduction is devoted to the idea and possibility of growing plants with higher nutritional quality in PFAL by using advantages of spectral modulability etc. Indeed, leafy vegetables are recognised as a source of minerals. However, nutritional quality (in this case mineral content) is usually measured in mg/100 gFW. Therefore in this paper CL-treated lettuce plants are not enriched with minerals as their content is lower than in NL-treated plants. In this case it is not possible to claim such plants to be functional vegetables. The accumulation values (calculated as mineral content per plant) do not play any role in this respect. In the abstract (line 38) it is noted, that CL treatment resulted in plants with high K, Ca, Fe and Zn contents, which is not true as follows from the results (Fig. 3a).
The data that lettuce plants gain greater shoot fresh weight and dry matter under CL (200 µmol m-2 s-1) than under normal photoperiod) have already been published (Zha et al., 2019) and do not add novelty to the paper. Therefore, the focus is to be made on nutrient content of CL-treated plants. But they are not to be called health-beneficial. We observe typical example of a dilution response whereby essential mineral-element concentrations in shoots would decrease as shoot-dry-matter accumulation increased.
Author Response
Responses to reviewer 2
First of all, we want to thank the reviewer for your helpful comments. We had revised the manuscript thoroughly according to reviewer comments.
- Truly, nutritional quality of leafy vegetables (in this case mineral content) is usually evaluated in fresh base. However, they can only be determined in dry shoot sample. In this study, if necessary, the data can be changed into fresh weight base. I think current data do give us some helpful and scientific results.
- The published paper "Morphological and Physiological Stress Responses of Lettuce to Different Intensities of Continuous Light" was cited.
Reviewer 3 Report
File attached

Author Response
Responses to reviewer 3
First of all, we want to thank the reviewer for your helpful comments. We had revised the manuscript thoroughly according to reviewer comments.
- Abstract is rewritten, and the introduction was revised carefully.
- Introduction, material and method, result and discussion were revised substantially.
- All problems put forward in the comments for reviewer 3 were revised carefully.
Round 2
Reviewer 1 Report
The manuscript was revised according the main aspects of previous review. Overall quality is average, however, some important remarks are still remaining:
Title, abstract and whole text - English language style should be revised!
L45-46 Some elements both increased and decreased under the same CL:
CL reduced N, P, K, Ca, Mg, Cu and Zn contents in lettuce shoot, while Fe and Mn contents did not change. Moreover, CL increased Ca, Fe and Mn accumulation.
The same misunderstanding in conclusions, L 297-298
L50-52 Information is repeated two times
L 57 and elsewhere in the text. “Functional vegetable” I would discuss, if the degree of the increase of mineral elements is significant enough to treat these lettuces as functional vegetable.
Results (also abstract) - it is not enough to indicate “the highest” and the lowest contents of elements, plant biomass etc. It is important to indicate the extent of the differences between light treatments - in%, times, from concentration xx to concentration yy etc.
Same comment as in the primary review - in discussion, obtained results and reference works are analyzed separately, both parts should be connected together.
Author Response
Responses to reviewer 1
First of all, we do want to thank the reviewer for your careful and very helpful comments. We understand your comments and revised the manuscript thoroughly. All revisions were changed into red words.
- Comment: The manuscript was revised according the main aspects of previous review. Overall quality is average, however, some important remarks are still remaining: Title, abstract and whole text - English language style should be revised!
Answer: We had revised the abstract and whole text. If necessary, we will revise them more times.
- Comment: L45-46 Some elements both increased and decreased under the same CL: CL reduced N, P, K, Ca, Mg, Cu and Zn contents in lettuce shoot, while Fe and Mn contents did not change. Moreover, CL increased Ca, Fe and Mn accumulation.
Answer: It is true results draw from the data. Contents in lettuce shoot responded to CL compared to NL treatment. I revised the sentence by adding “compared to NL”.
- Comment: The same misunderstanding in conclusions, L 297-298
Answer: It is true results draw from the data.
- Comment: L50-52 Information is repeated two times
Answer: I deleted the sentence “In addition, N, K, Fe, Zn and Cu accumulation was similar for three light qualities”.
- Comment: L57 and elsewhere in the text. “Functional vegetable” I would discuss, if the degree of the increase of mineral elements is significant enough to treat these lettuces as functional vegetable.
Answer: I agree with you. Thank you for your helpful comment. I had deleted the words “functional vegetable”. The aim of this study is mainly to clarify the effects of CL on yield and nutrient uptake which may have significance for improving nutritional value of hydroponic lettuce.
- Comment: Results (also abstract) - it is not enough to indicate “the highest” and the lowest contents of elements, plant biomass etc. It is important to indicate the extent of the differences between light treatments - in%, times, from concentration xx to concentration yy etc.
Same comment as in the primary review - in discussion, obtained results and reference works are analyzed separately, both parts should be connected together.
Answer: I had revised the related sentence. The word “highest” was changed into “higher”. All related context was revised.

Reviewer 2 Report
My prevuous comment is not taken into account and probably is not understood as follows from the authors' response. The question is not dry or fresh mass to use. The matter of fact that 'accumulation' calculated as nutrient content multiply by dry shoot weight is not an indicator of nutritional quality. In fact, the results show that nutrient content per DM (N, P, K, Ca, Mg, Cu, Zn) in CL-treated plants is lower than in NL-treated plants. Therefore it is not true to speak about functional plants or plants with high-level health-beneficial mineral elements as authors do. Greater dry biomass ensures higher accumulation of elements, but not higher nutritional quality per 100 g of FM or DM. Once more, I insist on that results show typical example of a dilution response whereby essential mineral element concentrations in shoots would decrease as shoot dry matter accumulation increased.
When discussing the results of spectral quality experiment 'the highest' doesn't mean higher than control (NL), therefore not 'beneficial'.
Line 108 and in the list if references: write cottect latin name: Taraxacum mongolicum
NL - normal light means 16/8 h photoperiod according to the text and 12/12 h in the table. What photoperiod was used?
Author Response
Responses to reviewer 2
First of all, we do want to thank the reviewer for your careful and very helpful comments. We understand your comments and revised the manuscript thoroughly. All revisions were changed into red words.
- Comments: My previous comment is not taken into account and probably is not understood as follows from the authors' response. The question is not dry or fresh mass to use. The matter of fact that 'accumulation' calculated as nutrient content multiply by dry shoot weight is not an indicator of nutritional quality. In fact, the results show that nutrient content per DM (N, P, K, Ca, Mg, Cu, Zn) in CL-treated plants is lower than in NL-treated plants. Therefore it is not true to speak about functional plants or plants with high-level health-beneficial mineral elements as authors do. Greater dry biomass ensures higher accumulation of elements, but not higher nutritional quality per 100 g of FM or DM. Once more, I insist on that results show typical example of a dilution response whereby essential mineral element concentrations in shoots would decrease as shoot dry matter accumulation increased.
Answer: I agree to your comment and opinion.
First, nutrient element accumulation is not a quality indication as you put forward. However, the accumulation can be regarded as an indicator to evaluate total mineral element uptake.
Second, CL truly tends to reduce part mineral element contents compared to NL plants due to large dry biomass. However, Fe and Mn contents were similar between CL and NL plants. That means dilution effect does not explain all the data. In addition, among CL treatments with different intensities and qualities presented various effects on mineral element contents. Thus, the data showed us suitable CL has potential to cultivate hydroponic lettuce with higher shoot mineral element contents and dry biomass.
So, I revised the abstract substantially.
- Comments: When discussing the results of spectral quality experiment 'the highest' doesn't mean higher than control (NL), therefore not 'beneficial'.
Answer: Some sentences were revised to express clearly.
- Comments: Line 108 and in the list if references: write cottect latin name: Taraxacum mongolicum
Answer: The words were revised.
- Comments:
NL - normal light means 16/8 h photoperiod according to the text and 12/12 h in the table. What photoperiod was used?
Answer: The 12/12h in the table had been changed into 16/8h.

Reviewer 3 Report
The manuscript is about how different light intensities and spectral qualities affect the growth and nutrient element content in lettuce. This is revised version of the manuscript and it have been improved significantly as compared to previous version. I recommend acceptance of manuscript after minor revision. The specific comments on sections are mentioned below-
Line 134- Was this nutrient solution pH and EC. If so, how was it maintained because within hydroponics it changes very quickly. I think this citing reference should help you to show that recommended EC for lettuce production in hydroponics- Electrical Conductivity and pH Guide for Hydroponics (HLA-6722) by singh and dunn.
Line 155- Are these elements explained somewhere before in manuscript. Explain their full forms at first use
Author Response
Responses to reviewer 3
First of all, we do want to thank the reviewer for your careful and very helpful comments. We understand your comments and revised the manuscript. All revisions were changed into red words.
- Comment: The manuscript is about how different light intensities and spectral qualities affect the growth and nutrient element content in lettuce. This is revised version of the manuscript and it have been improved significantly as compared to previous version. I recommend acceptance of manuscript after minor revision. The specific comments on sections are mentioned below.
Answer: Thank you very much for your helpful comments.
- Comment: Line 134- Was this nutrient solution pH and EC. If so, how was it maintained because within hydroponics it changes very quickly. I think this citing reference should help you to show that recommended EC for lettuce production in hydroponics- Electrical Conductivity and pH Guide for Hydroponics (HLA-6722) by singh and dunn.
Answer: Thank you for your helpful information. I have downloaded and learned the document. In our study, we renewed the nutrient solution each week.
- Comment: Line 155- Are these elements explained somewhere before in manuscript. Explain their full forms at first use
Answer: Full name of all nutrient elements were added, i.e. carbon (C), nitrogen (N), phosphorus (P), potassium (K), calcium (Ca), magnesium (Mg), iron (Fe), manganese (Mn), copper (Cu) and zinc (Zn).
